# Preparation and Characterization of High Mechanical Strength Chitosan/Oxidized Tannic Acid Composite Film with Schiff Base and Hydrogen Bond Crosslinking

**DOI:** 10.3390/ijms23169284

**Published:** 2022-08-18

**Authors:** Zhiyong Qin, Youjia Huang, Siyu Xiao, Haoyu Zhang, Yunlong Lu, Kaijie Xu

**Affiliations:** 1School of Resources Environment and Materials, Guangxi University, Nanning 530004, China; 2Guangxi Key Laboratory of Processing for Non-Ferrous Metals and Featured Materials, Nanning 530004, China

**Keywords:** chitosan, laccase, oxidized tannic acid, Schiff base, mechanical properties

## Abstract

Chitosan-based composite films with good biodegradability, biocompatibility, and sustainability are extensively employed in the field of food packaging. In this study, novel chitosan/tannic acid (CTA) and chitosan/oxidized tannic acid (COTA) composite films with excellent mechanical and antibacterial properties were prepared using a tape casting method. The results showed that, when 20% tannic acid (TA) was added, the tensile strength of the CTA composite film was 80.7 MPa, which was 89.4% higher than that of the pure chitosan (CS) film. TA was oxidized to oxidized tannic acid (OTA) with laccase, and the phenolic hydroxyl groups were oxidized to an o-quinone structure. With the addition of OTA, a Schiff base reaction between the OTA and CS occurred, and a dual network structure consisting of a chemical bond and hydrogen bond was constructed, which further improved the mechanical properties. The tensile strength of 3% COTA composite film was increased by 97.2% compared to that of pure CS film. Furthermore, these CTA films with significant antibacterial effects against *Escherichia coli* (*E. coli*) are likely to find uses in food packaging applications.

## 1. Introduction

The over-exploitation of fossil resources has caused serious environmental problems and had irreversible negative impacts on the environment [1]. To reduce the use of petroleum fuels and achieve carbon neutrality, substitute resources and technology need to be developed [2]. Biomass materials, as green renewable materials with potential for sustainable development, show good prospects for the replacement of traditional petroleum fuels and related synthetic products [3]. Chitosan (CS), the second largest biomass material after cellulose on the earth, is highly sought after and widely employed because of its abundance of sources, biodegradability, and biological compatibility [4,5,6]. Chitosan-based composite materials, such as hydrogels, aerogels, and films, have been widely used in biomedicine, sewage treatment, and food packaging. However, due to the poor mechanical properties of CS film, its uses and development are limited [7].

Tannic acid (TA) is a natural polyphenol widely distributed in the shells, leaves, and pulp of several plants, such as pomegranate peel, tea, lacquer tree, and pearl nuts [8,9]. The chemical structure of TA is composed of gallic acid dimers [10], which can be used to improve the properties of proteins, CS, and other biological macromolecules [11]. With excellent therapeutic and repair capabilities, including anticancer, antibacterial, antioxidant, and wound-healing properties, as well as steady-state quality [12,13], it has been studied and applied in many fields related to drug transport, biosensors, antioxidants, and biosorption [14,15]. The catechol and pyrogallol functional groups of TA can interact with various substances thanks to a variety of features, including hydrogen bonds, π-π interactions, hydrophobicity, coordination action, and dynamic covalent interactions with borate ester groups [16], and the functional groups on TA can form a metal–TA supramolecular amorphous network by coordinating with various polyvalent metal ions [17].

Researchers have attempted to address the problem of the poor mechanical quality of CS films. Taheri et al. [18] prepared chitosan/gelatin films with the addition of TA. The mechanical properties and the wound-healing effectiveness of the films were investigated, and the films were found to be suitable for use as wound dressings. Films made of CS, gelatin, and TA were prepared by Zhang et al. [19]. The mechanical and UV resistance qualities were enhanced by the addition of TA, and the films showed potential for application in fresh-cut apple preservation and other fields. Aewsiri et al. [20] found that non-covalent bonds were formed by TA and gelatin when the pH value was 7, while oxidized tannic acid (OTA) combined with gelatin through covalent bonds at the pH value of 9. Therefore, OTA might be a better crosslinking agent than TA and can be used to improve the mechanical properties of films.

Laccase is a kind of polyphenol oxidase containing copper [21] that was first discovered by Japanese scholar Yoshi in lacquer tree [22], and it is widely present in nature, including in fungi, plants, insects, animal viscera, and blood [23,24]. Numerous polyphenols, polyamines, aromatic amines, and aromatic thiols can be oxidized by laccase [25,26,27]. A variety of compounds (phenols, polyphenols, aniline, lignin) can be oxidized by laccase into radicals and only produce water as a by-product [27,28]. Lignocellulose materials can achieve 80.13% delignification under the catalysis of laccase, which has great potential for the preparation of bioethanol [29]. In the study by Bozic [30], caffeic acid, and gallic acid were used in CS crosslinking under the catalysis of laccase to prepare CS with good antibacterial and antioxidant properties. Under the catalysis of laccase, polyvinyl alcohol (PVA) has been crosslinked with ferulic acid to prepare hydrogels, which showed potential application value in biomedicine [31].

In this study, OTA was obtained by laccase oxidation of phenolic hydroxyl groups on TA and, through the reaction of OTA with amino groups on CS, a synergistic network consisting of the chemical bond and hydrogen bond was constructed. Novel chitosan/tannic acid (CTA) and chitosan/oxidized tannic acid (COTA) composite films with excellent mechanical and antibacterial properties were prepared using the tape casting method, which is the most commonly used method for preparing biomass films. To comprehensively assess the qualities of CS composite films, the mechanical, antibacterial, UV barrier, and antioxidant properties were investigated, and the mechanism underlying the films made of CS and TA crosslinked through a Schiff base reaction and hydrogen bond is explained.

## 2. Results and Discussion

### 2.1. Fourier-Transform Infrared (FTIR) Analysis

Figure 1 shows the FTIR test results for the TA and OTA powders. In comparison to the FTIR curves for TA, the peak intensities of OTA at 3200–3700 and 1325 cm^−1^ decreased, which was related to the oxidation of phenolic hydroxyl groups on TA to the o-quinone structure by laccase. The absorption peaks at 1198 and 1536 cm^−1^ may have been due to the stretching vibration of carbonyl (C=O), indicating that there were ester groups in OTA. In addition, the characteristic peak of the o-quinone structure appeared at 1647 cm^−1^, which indicated that the phenolic hydroxyl of the TA structure had been oxidized to active o-quinone under the action of laccase [32,33].

The FTIR results for the CTA and COTA films are shown in Figure 2. Figure 2a shows that the hydroxyl characteristic peak signal at 3360 cm^−1^ was enhanced after adding TA, which was due to the presence of many phenolic hydroxyls (-OH) in the TA. At 2920 cm^−1^, the signal strength of the characteristic peak was increased and shifted from 2918 cm^−1^ in the pure CS film to 2922 cm^−1^, signifying that TA was successfully introduced into the structure of the CS. The peak signals at 1540 and 1630 cm^−1^ were enhanced and shifted, which could be attributable to the stretching vibration caused by C=C in TA. In Figure 2b, compared with the CS film, the hydroxyl peak of the COTA film at 3362 cm^−1^ showed an enhancement effect, signifying that OTA was successfully introduced into the structure of the CS. The peaks near 1630 cm^−1^ shifted to 1642 and 1645 cm^−1^, which was related to the C=N bonds generated by the Schiff base reaction between the active o-quinone structure in the OTA and the CS amino group [34]. 

### 2.2. X-ray Diffraction (XRD) Analysis

The XRD results for the CS, CTA, and COTA composite films are shown in Figure 3. There were three distinct crystallization peaks in the CS films at 2θ of 10.9, 17.7, and 22.2°, respectively [35]. The peak at 2θ of 22.2° was attributed to the regular lattice of the CS molecules, and the peak at 2θ of 10.9° corresponded to the combination of water molecules in the lattice [36]. As shown in Figure 3a, the crystallization peaks at 2θ of 22.2°, 17.7°, and 10.9° were still retained, signifying that the CTA film had good chemical compatibility [37]. On the other hand, there was a certain interaction between the hydroxyl groups in the TA and CS films [38]. As shown in Figure 3b, compared with pure CS films, the peak strength of the COTA films decreased. This phenomenon indicates that the pure CS films themselves were a short-range ordered structure, and the addition of OTA affected the formation of the internal lattice structure of the composite film. Moreover, the benzene ring structure in OTA also affected the lattice structure [39].

### 2.3. X-ray Photoelectron Spectroscopic (XPS) Analysis

The XPS results for the CS, CTA-2, and COTA-1 composite films are shown in Figure 4 and Figure 5. As shown in Figure 4a, the peaks at 284.8 eV and 531.8 eV represented C 1s and O 1s, respectively. After TA was added, the O 1s content of the CTA-2 composite film at 531.8 eV was significantly higher than that of the pure CS film, which was caused by the presence of the O element in TA. After the addition of OTA, the content of O elements in COTA-1 decreased compared to CTA-2, signifying that the o-quinone structure in the OTA reacted with the amino group in the CS molecular chain and some of the water molecules were removed. Figure 4b,c reveal that the signal intensity of the C 1s peak in the CTA-2 composite film was increased due to the presence of many C-C, C=C, C=O, and C-O groups in the TA structure. According to Figure 4d, after the addition of OTA, the peak intensity of the COTA-1 composite film at 287.8 eV was higher than that of the pure CS film, and the peak intensity near 286.2 eV decreased. This was because the Schiff base reaction occurred between CS and OTA, which transformed some C=O bonds into C=N bonds and removed water molecules. The N1s test results for the composite film are shown in Figure 5a–d. In contrast to the CS and CTA-2 films, the C=N peak appeared at 401.3 eV in the COTA-1 film, which also confirmed that the Schiff base reaction occurred between CS and OTA molecules.

### 2.4. Thermogravimetric Analysis (TGA)

The TGA results for the CS and CTA composite films are shown in Figure 6a,b and Table 1. The decomposition temperatures corresponding to the maximum decomposition rates in the CS and CTA-4 films were 284.0 °C and 282.4 °C, respectively. The final residual rate in the composite films increased because the strong hydrogen bonding between CS molecules was destroyed by TA, and the benzene ring structure of TA also affected the hydrogen bonding between the TA and CS molecules [40]. The residual percentage for the CS film in the final 600 °C was 38.98%, and the residual percentage for the CTA-4 film after adding 20% TA increased to 42.01%. A possible explanation for this result could be that the existence of benzene rings affected the internal network structure through spatial conformation, increasing the thermal degradation residual rate for the CTA composite films.

As shown in Figure 6c,d and Table 1, the final residual ratio for the CS film was 38.98%, and the final residual ratios for the OTA composite film were 36.98%, 36.59%, and 35.62%, respectively. The maximum degradation temperature for COTA-3 dropped to 281.4 °C from 284.0 °C for the CS film. This reveals that OTA reacted with CS to form a chemical bond network, which destroyed the original hydrogen bond network and led to the reorganization of the internal structure, thereby reducing the maximum degradation temperature.

### 2.5. UV Absorption Spectra, Fluorescence Properties, and Color Aberration Analysis

As shown in Figure 7a, at 365 nm, the transmittance of the CS solution was 59.41%. The transmittance of the CTA-1, CTA-2, CTA-3, and CTA-4 solutions at 365 nm were 16.48%, 14.75%, 11.66%, and 11.46%, decreasing by 72.26%, 75.17%, 80.37%, and 80.71%, respectively. The transmittance values for CTA-3 and CTA-4 were close, indicating that the improvement in the UV barrier performance was limited by adding more TA, and the phenolic hydroxyl groups and chromophores in TA absorbed ultraviolet light, resulting in a decrease in the transmittance at 365 nm [19]. As shown in Figure 7b, the UV barrier properties of COTA were better than those of the CS film solution at 365 nm. The UV transmittance values for the CS, COTA-1, COTA-2, and COTA-3 solutions at 365 nm were 59.39%, 56.05%, 55.65%, and 55.39%, respectively. The transmittance values for the COTA composite film solution decreased by 5.66%, 6.33%, and 6.77% compared to CS. 

Due to its internal structure, CS powder and film can emit light blue fluorescence under UV excitation at 365 nm [41]. Figure 7d,f show the fluorescence images of the CS, CTA, and COTA films under 365 nm UV excitation. As shown in Figure 7d, the CS solution showed a strong fluorescence effect, while the fluorescence of the CTA composite films nearly disappeared under 365 nm UV illumination. The reason for the weakening of the fluorescence was that the benzene ring structure in TA affected the internal structure of the film through spatial conformation. As can be seen in Figure 7f, the fluorescence of the OTA solution was slightly weaker than that of the CS solution. Under 365 nm UV light irradiation, the fluorescence color of the COTA composite film was close to that of the CS film, but the fluorescence intensity was weaker than that of the CS film. This implies that the chemical structure of OTA changed compared to TA, the terminal part of the phenolic hydroxyl was oxidized to an o-quinone structure, and the Schiff base reaction between OTA and CS formed a C=N bond.

The color aberration results for the CTA and COTA composite films are shown in Table 2. The L* value, which denotes brightness, ranges from 0 to 100, and the a* value describes the intensity of the red and green colors. Positive a* values represent red, while negative a* values represent green. The b* value indicates the intensity of the yellow and blue colors. A positive b* value indicates yellow, while a negative b* value indicates blue. Figure 7c,e show images of the CS, CTA, and COTA films under visible light. As indicated in combination with Table 2, the colors of the CTA and COTA films increased with the increase in TA. The increase in the OTA ratio led to a deepening of the colors of the composite films. This result was consistent with the images shown in Figure 7c,e.

### 2.6. Mechanical Properties 

Figure 8 shows the tensile properties of the CTA and COTA composite films. The narrowest width in the CTA and COTA films was 2 mm. According to the stress–strain curves (Figure 8a), the tensile strength of the CS film was 42.6 MPa, and the elongation at break was 23.7%. After adding different proportions of TA, the tensile strength of the CTA composite films increased from 42.6 MPa to 54.6 MPa, 58.8 MPa, 73.5 MPa, and 80.7 MPa, increasing by 28.2%, 38.0%, 72.5%, and 89.4%, respectively. The reason was that there were phenolic hydroxyl groups in the TA molecules, forming a dense hydrogen chain network between the CS molecules and improving the tensile strength. The hydrogen bond network and crystal structure inside the CS were destroyed by the TA. The tensile strength and elongation at break of the CTA-4 composite film were 80.7 MPa and 5.6%, respectively. The benzene ring in the TA structure limited the movement and stretching of the chain through spatial conformation, which was an important reason for the high tensile strength and low elongation at break of the CTA-4 film.

Table 3 summarizes the specific test results. After adding OTA, the tensile strength values for the COTA composite films were 68.4 MPa, 72.4 MPa, and 84.0 MPa. Compared to the CS films, the tensile strength of the COTA composite films increased by 60.6%, 70.0%, and 97.2%, respectively. This indicated that, with the addition of OTA, the o-quinone structure in OTA reacted with the amino group in CS to form a chemical bond. As shown in Figure 1, when the tensile fracture occurred, the CS molecular chain was the first to be acted upon, being stretched from a curling state to a long, straight chain state and, subsequently, it was gradually broken. Finally, the hydrogen bond network and chemical bond network inside COTA were destroyed in turn. The results in Figure 8b reveal that the strain in the COTA films was reduced compared to that of the pure CS film. These changes are likely attributable to the increase in the tensile strength of the COTA composite film, which was caused by the synergistic effect of the internal hydrogen bonds and chemical bonds. 

### 2.7. Microstructure Analysis

In the next part of the study, the tensile fracture sections of the CS, CTA, and COTA films were examined using SEM to explore the changes in the microstructure of the CS films after crosslinking with OTA. The specific scanning images are shown in Figure 9, and the following figure shows that the tensile fracture surface of the CS film was relatively flat. When TA and OTA were crosslinked with the CS films, the fracture sections in the CTA and COTA composite films showed different rough surfaces and bulges.

In comparison to the CS films, the tensile fracture sections of the CTA composite films became rough, indicating that the presence of a benzene ring structure in the TA molecule limited the stretching and elongation of the chain and the rotation of the bond angle. The tensile fracture sections in the COTA-1, COTA-2, and COTA-3 composite films were dense, but there were some obvious bulges. This phenomenon indicated that chemical bonds were introduced into the composite films, and a dual network of chemical bonds and hydrogen bonds was successfully constructed inside the COTA composite films. As shown in Table 3, the elongation at break of the composite films decreased after TA was crosslinked with CS, and the elongation at break first increased and then decreased after crosslinking of OTA with CS. During the fracture process of the COTA composite films, the increase or decrease of the elongation at break led to the corresponding changes in the film, resulting in the obvious bulges in the film section [42]. 

The atomic force microscopy results for the CS, CTA-2, and COTA-1 films are shown in Figure 10. In this figure, the upper part shows the atomic force microscope results for the film, and the lower part shows the 3D imaging results for the film. The specific values for the film surface roughness are summarized in Table 4, where Sz represents the sum of the maximum peak height and the maximum valley depth, Sa represents the average surface relative to the surface, and Sq represents the standard deviation based on the average height. Results for the Sa of CS, CTA-2, and COTA-1 were 9.1 nm, 16.4 nm, and 9.4 nm, and the results for the Sq were 11.7 nm, 22 nm, and 11.9 nm, respectively. This demonstrates that the surface of the CTA-2 film was relatively rough and that the COTA-1 and CS films had similar surface roughness. During the film-forming process in the CTA composite film, partial TA aggregation occurred, resulting in poor uniformity in the film. The addition of OTA led to a reaction with CS that formed a chemical network, and the film structure was more uniform and denser. Similarly, the 3D imaging results for the film show that, after the addition of OTA, the bulges on the film surface were reduced and the film was smoother and more uniform. 

### 2.8. Solubility and Water Absorption Analysis

The solubility, water absorption, and water content of the CS-based composite films are summarized in Table 5. The thicknesses of the films with different compositions varied between 0.111 ± 0.002 and 0.160 ± 0.005 mm. The solubility of the pure CS film after 24 h immersion was 18.31%. The solubility of the CTA-1 composite film increased to 20.56%, indicating that the addition of TA weakened the strong hydrogen bond network in CS, which facilitated the entry and penetration of water molecules in the test process, leading to the increase in film solubility. When the added amount of TA increased, the solubility of the CTA composite film decreased compared to the solubility of the CTA-1 film. This reveals that amino groups in TA and CS molecular chains formed a denser hydrogen bond network, which enhanced the compactness of the film and reduced the probability of water molecules entering, resulting in the decrease in the solubility of the film. The solubility values for the COTA-1, COTA-2, and COTA-3 composite films were 17.53%, 10.0%, and 10.32%, respectively. A possible explanation for this effect is the Schiff base interaction between the amino group in CS and the o-quinone structure in OTA: the formation of chemical bonds resulted in a denser three-dimensional network structure inside the composite film, and then the solubility of the COTA composite film decreased with the increase in the OTA content.

### 2.9. Analysis of Antibacterial Activity and DPPH Radical Scavenging Rate

Figure 11 shows that the diameter of the inhibition zone of CS against *S. aureus* was 14.51 mm, and the diameters of the inhibition zones of the CTA films were 13.02 mm, 13.53 mm, 13.35 mm, and 12.87 mm, respectively. After adding OTA, the diameters of the inhibition zones of the COTA films were 14.06 mm, 13.55 mm, and 13.5 mm, respectively. The diameter of the inhibition zone of CS against *E. coli* was 14.56 mm, and the diameters of the inhibition zones of the CTA films were 14.63 mm, 14.49 mm, 16.09 mm, and 16.15 mm, respectively. After adding OTA, the diameters of the inhibition zones of the COTA films were 13.02 mm, 15.01 mm, and 15.04 mm, respectively. With the introduction of OTA, the diameters of the inhibition zones of the COTA films against *S. aureus* and *E. coli* remained the same. It is worth noting that the antibacterial effects of the CTA-3 and CTA-4 films against *E. coli* were obvious. 

The DPPH radical scavenging rates of the CS, CTA, and COTA composite films are shown in Figure 12. The radical scavenging rate of DPPH on the CS film was 37.26%, which resulted from the presence of a nitrogen atom in the amino group of CS [43]. After the addition of TA, the antioxidant activity of the CTA composite films was greatly improved, and the DPPH radical scavenging rates of CTA-1, CTA-2, CTA-3, and CTA-4 were 94.84%, 95.53%, 96.01%, and 96.11% compared with CS, increasing by 254.5%, 256.4%, 257.7%, and 257.9%, respectively. The main reason for the increase was that electrons were transferred from the phenolic hydroxyl of TA to radicals, which prevented the chain polymerization of radicals and further slowed down the oxidation reaction [44]. The DPPH radical scavenging rate of the COTA composite films was lower than that of the pure CS film but, with the increase in OTA ratio, the DPPH radical scavenging rate of the COTA composite film also increased, and the results showed that the DPPH radical scavenging rates of the COTA composite films were 31.23%, 32.59%, and 36.96%, respectively. This indicated that phenolic hydroxyl groups in the TA were oxidized to the o-quinone structure by laccase, reducing the presence of hydroxyl groups. One possible explanation for this effect is that the Schiff base reaction between the o-quinone structure and the CS molecules led to the formation of chemical bonds, forming a dual network structure of chemical bonds and hydrogen bonds and reducing the probability of phenolic hydroxyl groups providing electrons to radicals, thereby reducing the oxidation resistance of the COTA.

## 3. Materials and Methods

### 3.1. Materials

CS (deacetylation ≤ 90%, viscosity 100–200 mpa.s), TA (>99.5%), laccase (from *Coriolus Versicolor* with activity of 0.5 U/mg), sodium dihydrogen phosphate (99.99% metal basis), disodium hydrogen phosphate dodecahydrate (99.99% metal basis), glacial acetic acid(>99.5%), 2,2-biphenyl-1-picrylhydrazyl (DPPH, 96%), and Mueller-Hinton agar (MHA) were purchased from Shanghai McLin Biotech Co., Ltd. *Escherichia coli ATCC 25922* (*E. coli*) and *Staphylococcus aureus ATCC 25923* (*S. aureus*) were provided by Huankai Microbiological Technology Co., Ltd.

### 3.2. Preparation of OTA

First, 1 g of TA was added to 50 mL of a phosphate buffer solution with a pH of 6.5. After dissolution, 100 mg laccase was added and reacted at room temperature for 15 min. The reaction mixture was then transferred to a water bath at 30 °C for 24 h to obtain the OTA. The obtained solution was centrifuged at 12,000 r/h for 15 min and rinsed three times with deionized water. Then, the product was freeze-dried for 48 h to obtain the OTA powder.

### 3.3. Preparation of CTA and COTA Composite Films

The preparation of the CTA and COTA composite films is shown in Figure 2. Briefly, 1 g CS powder, 1 g acetic acid, and 98 g deionized water were mixed and stirred at room temperature. Different masses of TA (equivalent to 0, 5, 10, 15, and 20 wt% CS) were added to the CS solution, respectively, and the samples were correspondingly named CS, CTA-1, CTA-2, CTA-3, and CTA-4. Then, the samples were magnetically stirred in a water bath at 50 °C for 1 h and at room temperature for 12 h to form homogeneous CTA mixed solutions. Subsequently, the solutions were transferred to Petri dishes (Φ = 12 cm) and dried in a humidity box at a constant temperature of 45 °C and 50% RH.

The preparation of COTA films was similar to the previously mentioned CTA film preparation process. First, various concentrations of OTA (equivalent to 1, 2, and 3 wt% CS) were added to the CS solution, respectively, and the samples were correspondingly named COTA-1, COTA-2, and COTA-3. Then, the samples were heated and magnetically stirred in a water bath at 50 °C for 1 h and at room temperature for 12 h. Finally, they were transferred to Petri dishes (Φ = 12 cm) and placed in a humidity box at a constant temperature of 45 °C and 50% RH until the films were dried. The composition ratios of the CS, TA, and OTA composite films are shown in Table 6.

### 3.4. Characterization and Testing of Chitosan-Based Composite Film

#### 3.4.1. Fourier-Transform Infrared Spectroscopy (FTIR)

A Fourier-transform infrared spectrometer (IRTracer-100, Japan) was used to investigate the functional groups of the CTA and COTA composite films and the TA and OTA powders. The data were collected in 32 scans with a resolution of 4 cm^−1^ within the range of the wavenumbers from 4000 to 400 cm^−1^. The ATR-FTIR spectra were rectified through advanced ATR correction in OMNIC software.

#### 3.4.2. Scanning Electron Microscope (SEM)

A scanning electron microscope (Sigma 300, Germany) was used to observe the microstructures of the tensile failure sections of the CTA and COTA composite films. The samples were sprayed with gold before detection, and the detection voltage was 3 kV.

#### 3.4.3. Atomic Force Microscopy (AFM)

The surface roughness of the composite film was examined using an NX10 atomic force microscope (Park, South Korea), and it was subsequently observed using three-dimensional topography. The film was cut with a width of 5 mm, and the height and phase images of the films were captured on camera using a probe in tapping mode (resonance frequency between 250 and 300 kHz). Every two to four locations on each specimen were recorded.

#### 3.4.4. X-ray Photoelectron Spectroscopy Analysis (XPS)

The chemical element compositions and chemical bond states of the samples were analyzed using X-ray photoelectron spectroscopy (Thermo Scientific K-Alpha, UK). AlKα radiation (6 eV) with a monochromator as a radiation source at 100 W (12 kV working voltage, 6 mA filament current) was used, the vacuum of the sample bin was higher than 5.0 E-7mBar, and a binding energy of C 1s = 284.80 eV was the energy standard.

#### 3.4.5. Thermogravimetric Analysis (TGA)

A thermogravimetric analyzer (DTG-60, Japan) was used to test the thermal stability properties of the film samples. The film samples (3–5 mg) were tested in a nitrogen environment at a heating rate of 10 °C/min from 30 to 600 °C. The trend for the change in the sample mass with temperature was analyzed, and the maximum decomposition temperature for each sample was recorded.

#### 3.4.6. Mechanical Properties

An electronic universal tensile testing machine (ZQ990, Dongguan) was used for the tensile test. The CTA and COTA composite films were cut by a dumbbell-shaped cutter with a width of 25 × 2 mm. The test conditions were as follows: the original standard distance of the tensile test was 18 mm and the tensile speed was 2 mm/min. Then, the tensile stress, tensile strain, and Young’s modulus of the film were recorded. Three parallel samples were tested in each group, and the results were the average of the three test results. The tests were undertaken at 25 ± 1 °C, 50% RH.

#### 3.4.7. Ultraviolet–Visible (UV–Vis) Spectroscopy

The UV absorption and transmittance of the CTA and COTA composite film solutions were tested with a UV–Vis spectrophotometer (UV-1800PC, Shanghai) at 365 nm. Pure water was used as the control group, and the CS, CTA, and COTA solutions were poured into the cuvette for the UV absorption and transmittance tests, respectively. A measured sample solution was needed to ensure a uniform concentration.

#### 3.4.8. Fluorescence Spectrometer Analysis

A fluorescence spectrometer (HORIBA FluoroMax-4, France) was used to test the fluorescence spectrum performance of the CTA and COTA composite film solutions. The CS, CTA, and COTA solutions were poured into cuvettes. Then, the cuvettes were placed in the detection instrument. The excitation wavelength was set to 365 nm, the scanning wavelength was 300–800 nm, and the scanning speed was 5 nm/s. A measured solution was needed to ensure a uniform concentration.

#### 3.4.9. Antibacterial Properties

Single colonies of *S. aureus* and *E. coli* were resuscitated in Luria–Bertani (LB) medium, the bacteria were diluted to 10^7^ CFU/mL with 0.3 mL phosphate buffered saline (pH 6.8), and the 100 μL diluted bacterial solution was evenly spread on the surface of LB medium with a glass rod. Then, 200 μL of CS, CTA, or COTA solution was injected into the beaker with a pipette. After incubation at 37 °C for 12 h, the inhibition zone was measured with Vernier calipers in mm units, and each sample was measured on average three times.

#### 3.4.10. Water Solubility and Water Absorption

Six 15 × 15 mm samples were dried to constant weight (recorded as $m_{0}$) in an oven at 105 °C for 24 h and then immersed in deionized water for 24 h at room temperature. The insoluble film samples were further dried at 105 °C to a constant weight (recorded in $m_{d}$) [45]. The solubility of the composite film in water was calculated according to Equation (1):(1)$$W_{s}\left(\%\right)=\frac{m_{0}-m_{d}}{m_{0}}$$
where $W_{s}$ is the water solubility of the composite film (%), $m_{0}$ is the initial dry weight of the composite film before swelling (g), and $m_{d}$ is the absolute dry weight of the composite film after swelling (g).

Six 15 × 15 mm samples were used for the water absorption test. The samples were placed in a dryer at 25 °C and 95% RH and the initial weight (recorded as $m_{1}$) was recorded. The content of the distilled water was conditioned to maintain 92% relative humidity at room temperature (25 ± 2 °C). The films were removed after 48 h and weighed (recorded in $m_{t}$) [46]. The solubility of the composite film in water was calculated according to Equation (2):(2)$$W_{u}\left(\%\right)=\frac{m_{t}-m_{1}}{m_{1}}$$
where $W_{u}$ is the water absorption of the composite film (%), $m_{1}$ is the dry weight of the composite film before water absorption (g), and $m_{t}$ is the quality of the composite films at different times after water absorption (g).

#### 3.4.11. Color Aberration Analysis

The Hunter color (L*, a*, b*) values of the films were measured with a color analyzer (CA-210, Shenzhen). The film was cut into 30 × 30 mm samples, and white paper was used as the standard background of the measurement. Three different points on the film were measured, and the average value of the three points was used as the Hunter color value. The total color aberration (∆E) was calculated with the following equation:(3)$$\Delta E={\left[{\left(\Delta L\right)}^{2}+{\left(\Delta a\right)}^{2}+{\left(\Delta b\right)}^{2}\right]}^{\frac{1}{2}}$$
where ΔL represents light and dark changes, Δa represents red and green changes, and Δb represents yellow and blue changes.

#### 3.4.12. Scavenging Rate of the DPPH Radical

For this test, 1 mL of composite film solution was reacted with 4 mL of 100 μmol/L DPPH methanol solution at room temperature for 20 min to determine the absorbance of the reaction solution at 517 nm. The scavenging rate of the DPPH radical was calculated according to Equation (4):(4)$$R_{DPPH}\left(\%\right)=\frac{A_{1}-A_{2}}{A_{0}}$$
where $A_{1}$ is the absorbance of the DPPH methanol solution at 517 nm, $A_{2}$ is the absorbance of the reaction solution at 517 nm, and $A_{0}$ is the absorbance of DPPH aqueous solution at 517 nm.

## 4. Conclusions

In this study, CTA and COTA composite films were prepared using a tape casting method. The composite films had good UV resistance properties and antibacterial properties and excellent mechanical properties. Compared with the CS film, the tensile strength of the CTA composite film increased by 89.4% and the transmittance of the composite film decreased by 80.71%. Through the oxidation of TA by laccase, terminal phenolic hydroxyl groups in the TA were oxidized to an o-quinone structure. The o-quinone structure junction in the OTA reacted with the amino group in the CS molecule to form a chemical bond, resulting in the construction of a dual network structure consisting of a chemical bond and hydrogen bond that further improved the mechanical properties. The tensile strength of the COTA-3 film was 84.0 MPa, which was higher than the strength of 80.7 MPa for the CTA-4 film, indicating that the composite film with a dual network structure had more advantages than a single network structure. Compared with the CS solution, the UV resistance of the COTA composite film solution increased by 6.77%. The addition of TA improved its antibacterial effect against E. coli. Hence, the dual network structure consisting of a chemical bond and hydrogen bond simultaneously reinforced and toughened the pure CS films and could be used as a feasible and novel crosslinking method.

## Data Availability

Not applicable.

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
