# Peer review of "Preparation and Characterization of High Mechanical Strength Chitosan/Oxidized Tannic Acid Composite Film with Schiff Base and Hydrogen Bond Crosslinking"

_ijms, 2022, doi:10.3390/ijms23169284_

Round 1
Reviewer 1 Report
The authors have presented a sound study on the development and characterization of CTA and COTA films. The english as well as the scientific writing must be thoroughly revised as it is hard to keep track on the explanations Just some minor changes:
Intro: you are not mentioning much about the tape casting method in the intro, how is it relevant to other studies? is this method better than other film formation methods?
Methods
Table1- Table 1
section 2.4.4 space between analysis and XPS
section 2.4.7 what lambda are you measuring?
Section 3.6- should change the title to "mechanical properties"
Table 3-it is elongation at break, not breaking elongation.
Is there any way you can merge Fig 7 o 8 together? there are too many figures, normally a research article should have max 10 Figures, it looks too dense otherwise.
line 384-change indicated to indicating. From line 384 to 391, i could not understand much. The sentence is too long and it is not explained properly, please reword. Meantime does not make sense in this phrase
line 391- indicates instead of indicating. Reword the sentence that comes afterwards, it is too long and the english is poor
Table 5: add a space in the title "water content (%)". The thickness of the films are quite variable (i.e. 111 um vs 160 or 190 um). Won't this have an effect as well on the water absorption, solubility, etc?
Section 3.9 you mention throughout the text that S. aureus has little effect, but in the picture the halo sizes look similar in both strains. why is that? Also the graphs in Fig 13 are blurry, perhaps you could increase quality on this? also, from this graphs it doesnt look like inhibition zone dimaeter is that different?
Reviewer 2 Report
Preparation and Characterization of High Mechanical Strength Chitosan/Oxidized Tannic Acid Composite Film with Schiff Base and Hydrogen Bond Crosslinking
The authors conducted a very intensive studies on comparison of Tannic acid and oxidized Tannic acid on the properties of the Chitosan composite film with respect to the chemical interaction, thermal stability, water absorption/solubility, antibacterial effects, absorbance and mechanical. These are my comments.
1. Define the abbreviation of TA in the first it was appeared (line 39).
2. In the methodology, kindly differentiate the preparation of CTA and COTA since the statement in the methodology is very confusing. It is not clear whether there is a separate preparation for CTA and COTA (line 95 to 109). Kindly indicate also in the schematic representation.
3. The methodology of water solubility and water absorption (line 174-189). To determine the water absorption, is it the sample taken after water solubility test? It means that the samples remain from the solubility test were used for the determination of water absorption (line 183). Kindly clarify the two methods.
4. Improve the Figure 2a and b by eliminating the area where it is not necessary to explain and make the area of 1740cm- to 1545cm- to emphasize what were explained in the discussion.
5. XRD results (line 236 to 250). It seems that the discussion that the peak of CTA composite film at 2q of 2.22 and the peak at 2q of 10.9 shifted to the left and becomes more obvious with the increase of TA addition which is similar to COTA film. It seems from the XRD diffractograms, that there is no difference with the CS whether with addition of TA or even OTA. Please clarify the XRD results.
6. TGA – create a Table of TGA for CTA and COTA. Otherwise, indicate the decomposition temperature and residue in the graphs.
7. UV absorption spectra, Fluorescence properties and color aberration analysis – Explain first what is Figure 7a and b, all about before getting into the explanation in Fig.7c. Similar in Fig. d and e before Fig. 7f. Why is it in CTA, that there is decreased in 72.26%, 75.17%, 80.37% and 80.71% for the different TA concentrations while in COTA, there is an increase by 5.66%, 6.33% and 6.77% compared with CS. Please clarify this statement. Are these data are taken from transmittance or UV barrier properties? Can you say that the transmittance is equivalent to barrier properties? Please clarify with this. Indicate also what is Figure 8 a and b, c and d in the discussion for clear overview of the sample. What is the importance of knowing the UV absorption, fluorescence and color aberration of the composite film? What specific application of composite films. Is it necessary the said tests?
8. Figure 9 seems not the correct figure for the given discussion on line 338-339. Figure 9 shows the schematic diagram of the internal structure of CTA and COTA film while in the line 338-339 explains the color of the composite film.
9. Discuss the stress and strain curve on the effect of addition of TA and OTA. What are the pictures below the stress-strain curve indicate? Any significant of the photos?
10. Microstructure analysis – Explain the surface morphology of the sample images in terms of roughness, appearance, voids, etc. Why? Correlate the morphology in the results of tensile properties and chemical structures.
11. AFM. Explain the 3D of the AFM results.
12. Table 5 – the solubility, moisture content and water uptake. The water content as indicated in the Table was the initial water content of the film or did you measure it prior to water solubility test and water absorption. What these data indicates to the application?
13. Antibacterial activity and DPPH radical scavenging rate – Explain what is Fig. 13a and b, and Fig 14a and b.
